# SEMI-AUTOREGRESSIVE DECODING FOR EFFICIENT LLM INFERENCE

## ABSTRACT

Inference in large language models (LLMs) is often slow due to their autoregressive nature. In this work, we formulate a semi-autoregressive decoding paradigm for LLMs that delegates part of the expensive computation from the original large model to a smaller, more efficient autoregressive model. The core of our design lies in the separate modeling of token dependencies, where the large model handles long-term dependencies on distant tokens, while the smaller model addresses short-term dependencies on recent tokens. When employed as a draft model in speculative decoding, our method allows for substantial reuse of computation in the LLM without missing any token dependencies, thereby striking a good balance between draft quality and drafting speed. Experiments on text summarization, medical QA, code generation, and mathematical reasoning tasks demonstrates the efficacy of our method.

## 1 INTRODUCTION

Large Language Models (LLMs) have demonstrated remarkable capabilities across diverse natural language processing tasks (OpenAI, 2024; Dubey et al., 2024). However, their deployment in real-world applications is often hindered by the substantial inference latency. The autoregressive nature of LLMs exacerbates this issue, as generating $n$ tokens requires $n$ sequential passes through the model, each of which involves expensive computation of stacked attention layers (Vaswani, 2017).

To address the latency challenges, several approaches (Ding et al., 2024; Jiang et al., 2024; Leviathan et al., 2023) have been introduced to improve LLM inference efficiency by delegating parts of the computation to smaller models. Among these innovations, speculative decoding (Leviathan et al., 2023) has emerged as a particularly promising technique. This method leverages the observation that certain tokens within the same inference run are easier to predict and can be handled by a smaller model. Speculative decoding employs a smaller draft model to generate draft tokens, which are then verified and refined by the larger target model. Compared to other approaches, speculative decoding offers a distinct advantage by guaranteeing generation quality, as it always falls back to the target model if necessary. Since its introduction, speculative decoding has proven effective across a wide range of generation tasks and has become a widely adopted tool for efficient LLM inference.

The acceleration achieved by speculative decoding, however, hinges on two factors: (i) the *acceptance rate of the draft tokens*, as well as (ii) the *latency of the draft model itself*. As such, a key ingredient in speculative decoding has been the design of the draft model. Existing work has explored several designs of the draft model, including the use of a separate transformer model (Leviathan et al., 2023; Chen et al., 2023; Kim et al., 2024), training additional modules to predict multiple tokens simultaneously (Cai et al., 2024; Luk et al., 2024; Bhendawade et al., 2024), or leveraging selected layers of the target model itself as the draft model (Elhoushi et al., 2024; Bae et al., 2023; Zhang et al., 2024b). Despite reported successes, these designs often involve a trade-off between acceptance rate and latency: some designs achieve a high acceptance rate but with compromises on latency (Elhoushi et al., 2024; Bae et al., 2023; Zhang et al., 2024b; Leviathan et al., 2023), whereas others may offer low latency but at the expense of draft quality (Cai et al., 2024; Luk et al., 2024; Bhendawade et al., 2024; Fu et al., 2024; Gloeckle et al., 2024). The reason for this trade-off stems from the fact that proposing good draft tokens requires the draft model to be complex enough to competently process token dependencies, which in turn leads to considerable drafting latency.

Figure 1: Overview of the pipeline of semi-autoregressive decoding. Red: tokens generated and verified by the LLM. Green: draft tokens generated by the semi-autoregressive draft model. $\triangle$: the representation of the current sentence computed by the LLM. The draft model generates the draft token in an autoregressive way conditioned on the representation $\triangle$ computed by the LLM.

In this work, we propose a new draft model design that achieves both a high acceptance rate as well as low inference latency. We begin by introducing a unified probabilistic modeling framework to analyze the trade-off between the two metrics both conceptually and empirically — a key piece that is missing in existing literature. Through this analysis, we identify that the optimal trade-off hinges on the efficient and effective modeling of the dependencies over recent and distant tokens. Motivated by this insight, we develop a new draft design, *semi-autoregressive drafting*, which separately addresses the dependencies over distant and recent tokens of different-sized models. This approach effectively balances draft quality and latency, resulting in considerable acceleration. In summary, our main contributions are as follows.

- We propose a unified probabilistic framework for draft model design in speculative decoding, where we systematically compare their trade-off between acceptance rate and inference latency;

- Building upon this framework, we develop a new draft model that works in a semi-autoregressive fashion, serving as a high-quality yet computationally cheap draft model for speculative decoding. A systematic study in the trade-offs in different implementations of this model is also conducted;

- Focusing on downstream LLM applications (realized via fine-tuning after model pretraining), we validate our method across four text generation tasks with three models in two training modes, demonstrating its advantages in terms of acceleration, memory cost, and training convenience.

## 2 PRELIMINARIES

### 2.1 LARGE LANGUAGE MODEL

A large language model assigns a probability $p(\mathbf{w})$ to a sequence of words $\mathbf{w} = (w_1, ..., w_T)$. This joint probability is usually factorized using the chain rule:

$$p(\mathbf{w}) = \prod_{i=1}^{T} p(w_i | w_1, ..., w_{i-1}) \tag{1}$$

which reduces language modeling to the problem of estimating the conditional probability of the next word given the history of all preceding words, hence *autoregressive*. This autoregressive property imposes a sequential constraint on the decoding process, requiring $n$ sequential passes through the model to generate $n$ tokens, which limits the speed of generation.

### 2.2 SPECULATIVE DECODING

To accelerate inference in autoregressive language models, *speculative decoding* (Leviathan et al., 2023) leverages a more efficient approximation model, i.e. the draft model $M_q$, to generate $K$ draft tokens $\{d_1, ..., d_K\}$ from an approximate distribution $q(\mathbf{w}) \approx p(\mathbf{w})$. The larger target model $M_p$ is then used to validate or override the draft tokens.

Specifically, a draft token $w_i \sim q(w_i | w_{<i})$ is accepted if $q$ provides a sufficiently close approximation to the target distribution $p$. In this work, we enforce a greedy acceptance criterion: the draft token is accepted if $w_i = \arg\max_v p(v | w_{<i})$. Otherwise, the draft token is rejected, and the larger model overrides the draft by generating a new token from $w_i \sim p(w_i | w_{<i})$. If a draft token is rejected, all subsequent draft tokens are also discarded, and the generation process reverts to the newest token generated by $M_p$.

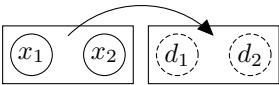 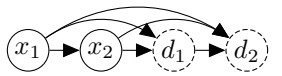 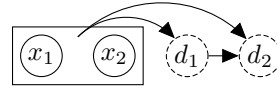

(a) Block autoregressive model     (b) Pruned autoregressive model     (c) Semi-autoregressive model

Figure 2: The graphical models of different draft model designs. *Non-dashed nodes*: tokens already generated and verified by the LLM. *Dashed nodes*: draft tokens to be generated by the draft model.

This process speeds up inference by exploiting parallelism in the draft token verification stage while retaining accuracy by falling back to the target model. Hence, the acceleration depends on the acceptance rate $\beta$, i.e. the probability of accepting $d_i$, as well as the latency of the draft model.

## 3 A UNIFIED FRAMEWORK FOR DRAFT MODEL

In this section, we present a unified view incorporating various draft model designs in speculative decoding as special cases, focusing on the different considerations for token dependency modeling. In the following, we use $q$ to denote the output distribution of draft model $M_q$, $h_p(\cdot)$ and $h_q(\cdot)$ to represent the hidden states of the last layer of the target model $M_p$ and the draft model $M_q$ computed for the input tokens, respectively.

**Drafting as sampling from** $q$. Consider a draft model $M_q$ in speculative decoding. Drafting the next $K$ tokens $\mathbf{d} = \{d_1, ...d_K\} \in \mathbb{N}^K$ can be seen as sampling from its output distribution $q$ conditioned on previously accepted tokens $\mathbf{w}_{\leq i}$:

$$\mathbf{d} \sim q(d_1, ...d_K|\mathbf{w}_{\leq i}) = \prod_{j=1}^{K} q_j(d_j|\mathbf{d}_{<j}, \mathbf{w}_{\leq i}) \tag{2}$$

To improve decoding efficiency, we usually consider three desirable properties when designing $M_q$: (i) high acceptance rate: distribution $q$ should well approximate the target distribution $p$ of the LLM; and (ii) fast drafting: sampling from $q$ should be faster than sampling from $p$; (iii) $q$ is a *low-dimensional* distribution, meaning that $K$ is small. The last property is important in our design.

One way to meet the aforementioned requirements is to focus on a powerful yet computationally cheap mechanism for modeling dependencies among the tokens $\mathbf{d}_{<j}, \mathbf{w}_{\leq i}$. Specifically, there are two distinct types of dependencies: (i) the dependencies on *distant tokens* $\mathbf{w}_{<i}$, i.e., the tokens already accepted and verified by $p$; and (ii) the dependencies on *recent tokens* $\mathbf{d}_{<j}$, i.e., the draft tokens sampled from $q$. In fact, many previous works can be categorized by how they handle these dependencies. For example:

- *Block autoregressive model* (Cai et al., 2024; Luk et al., 2024; Gloeckle et al., 2024) focuses on maximizing fast drafting by ignoring the dependencies of recent tokens and parallelizing draft token inference. In this case, the draft tokens $d_k$ and $d_l$ are conditionally independent given $\mathbf{w}_{\leq i}$, i.e., $d_k \perp d_l | \mathbf{w}_{\leq i}$ (Fig. 2a), leading to:

$$q_j(d_j|\mathbf{d}_{<j}, \mathbf{w}_{\leq i}) = q_j(d_j|h_p(\mathbf{w}_{\leq i})) \tag{3}$$

where each $q_j(\cdot)$ is a different LM head implemented as an independent multilayer perceptron (MLP) and $h_p(\cdot)$ represents the hidden states of the last layer of the target model.

- *Pruned autoregressive model* (Elhoushi et al., 2024; Zhang et al., 2024b) aims to achieve a balance between fast drafting and high acceptance rate. Unlike the *block autoregressive model*, it proposes to model all token dependencies with a pruned version of the target model $M_p$, denoted as $M_{p'}$ (Fig. 2b):

$$q_j(d_j|\mathbf{d}_{<j}, \mathbf{w}_{\leq i}) = q_0(d_j|h_{p'}(\mathbf{d}_{<j}, \mathbf{w}_{\leq i})) \tag{4}$$

where $q_0(\cdot)$ is the LM head for the draft model and $h_{p'}(\cdot)$ is the hidden representation of a 'truncated' LLM, which can be either the first few attention layers of the target model $M_p$ (Elhoushi et al., 2024) or replacement of full attention with approximate attention (You et al., 2024). This pruned model can capture token dependencies and can be directly accelerated on commodity hardware depending on the type of pruning scheme employed.

**Summary.** The above two dependency modeling designs differ in terms of (a) acceptance rate and (b) drafting speed. In particular, *block autoregressive model* ignores the dependencies among *recent tokens* $\mathbf{d}_{<j}$, which enables the reuse of hidden states $h_p$ to achieve highly efficient parallel decoding. However, this simplification harms the acceptance rate. In contrast, *pruned autoregressive model* captures all token dependencies through a pruned LLM, aiming to balance fast sampling (through pruning) and high draft quality (through dependency modeling). However, its accuracy is highly dependent on the pruned model. Furthermore, it processes both recent and distant tokens together without distinguishing them, leading to a sub-optimal design. To address these issues, we propose a new design based on the unified view of the draft model. This approach improves the balance between drafting speed and accuracy by separately modeling the two types of token dependencies.

## 4 METHODOLOGY

### 4.1 HIGH-LEVEL DESIGN

The core intuition of our proposal is to combine the advantages of the two previous models, i.e., maximizing hidden state re-use while effectively modeling token dependencies. Specifically, we propose to model distant token and recent token dependencies separately, using the target model $M_p$ for distant tokens, and a tiny language model[1] (TLM) $M_q$ for recent tokens.

- *Distant token dependency*. These tokens determine the context and the semantics for drafting subsequent tokens. For these tokens, we process them with the original LLM $M_p$, where we reuse the hidden state $h_p(w_1, ...w_i)$ computed by the LLM. This hidden state represents the LLM's understanding of the text context. Similar to the block autoregressive model, this context representation only needs to be computed once and is reused throughout the drafting process.
- *Recent token dependency*. These tokens correspond to the $K$-gram generated by the draft model. To reduce the cost, we use a small model to compute the hidden states $h_q(d_1, ..., d_{j-1})$. These hidden states encode the local status of the draft. Since $K$ is relatively small, a very simple model for $M_q$ suffices, which we refer to as a TLM.

By combining these two design choices, we can model recent token dependencies, as opposed to the block autoregressive model, while incurring small or negligible additional computation cost compared to the pruned autoregressive model, through the use of a TLM. This approach unifies the strengths of LLMs and TLMs for draft generation. While LLMs are powerful in understanding text semantics, they are known to be expensive to run. We thereby use them sparingly, calling them only when the context of the text is likely to have changed. In contrast, TLMs are much cheaper to run, though they struggle to parse complex semantics. We thereby only use them to handle short phrases and $K$-grams, conditioned on the semantic understanding provided by the LLM. In this way, we achieve both high efficiency and good draft quality.

This design leads to what we call the *semi-autoregressive draft model* (see Fig. 2c), which is formally defined as follow:

*Definition* 1. (*Semi-autoregressive draft model*). *Let* $\mathbf{d} = \{d_1, ..., d_K\}$ *be the draft tokens and* $\mathbf{w}_{\leq i} = \{w_1, ..., w_i\}$ *be the tokens generated and already verified by* $M_p$. *A semi-autoregressive draft model is a probabilistic model defined by the following probabilistic distribution*:

$$q(\mathbf{d}|\mathbf{w}_{\leq i}) = \prod_{j=1}^{K} q_0(d_j|h_p(\mathbf{w}_{\leq i}), h_q(\mathbf{d}_{<j})) \tag{5}$$

*where* $h_p(\cdot)$ *and* $h_q(\cdot)$ *are the hidden states of an LLM and that of a TLM, respectively.*

**Inference.** As discussed, sampling from the above model is highly efficient: while the computation of $h_p(\mathbf{w}_{\leq i})$ is expensive, it only needs to be computed once and can be reused for all draft tokens. On the other hand, $h_q(\mathbf{d}_{<j})$ must be recomputed for each $j$, but this is cheap due to the light-weighted nature of $M_q$, which normally requires even significantly less computation than a single decoding layer of a transformer.

---

[1]We use the term tiny language models here to highlight that these models are extremely cheap to run.

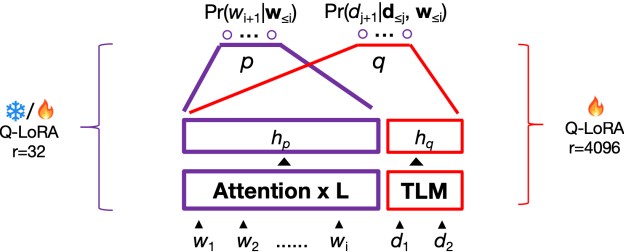

Figure 3: The architecture of the semi-autoregressive draft model, which separately models distant tokens $\{w_1, ... w_i\}$ and recent tokens $\{d_1, ... d_K\}$ with the hidden representations of the original LLM $h_p$ and a tiny language model (TLM) $h_q$. Note that $w_{i+1} = d_1$. The two models can be learned either jointly or separately using distinct Q-LoRA ranks.

## 4.2 IMPLEMENTATION

In this section, we discuss the detailed implementations of $M_q$ and the training procedure of the proposed model. Additional details on tree attention and KV caching can be found in Appendix A.

**Realization of $h_q$ and $q_0$.** We consider the following two implementations of the tiny language model $M_q$ for processing recent tokens $\mathbf{d}$. Both implementations are motivated by the fact that the number of recent tokens $K$ is small, so we only need to model short-range dependencies. Therefore, simple models are sufficient in this case, making our method significantly cheaper than conventional speculative decoding, which uses more expensive draft models.

- *Simplified transformer.* The first design is a one-layer transformer with hard-coded attention weights. Specifically, let $e(w)$ be the word embedding of a token $w$. We implement the network $h_q(\cdot)$ as follows:

$$h_q(\mathbf{d}_{\leq j}) = \frac{\sum_{l=1}^{j} \alpha_l \text{MLP}(e(d_l))}{\sum_{l=1}^{j} \alpha_l} \quad (6)$$

where $\boldsymbol{\alpha} = \{\alpha_1, ... \alpha_K\} \in (0, 1)^K$ are learnable parameters and MLP is a multi-layer perceptron with two hidden layers. While fixed attention weights may be inadequate for processing long context, it exhibits as an effective method for handling short texts (Raganato et al., 2020).

- *Recurrent networks.* The second design involves using an LSTM (Hochreiter, 1997) or even a vanilla RNN (Hochreiter, 1997) to model the local dependencies of the draft tokens[2]:

$$h_q(\mathbf{d}_{\leq j}) = \text{LSTM}(e(d_1), ..., e(d_j)) \quad (7)$$

The LSTM consists of two hidden layers, and its output is the hidden state of the last draft token. While LSTMs may struggle in long-context modeling due to their fixed-sized hidden state in contrast to transformers, they are well-suited to capture short-term dependencies.

The LM head $q_0$ is implemented as a simple MLP model with two hidden layers, and its input is the concatenation of $h_q$ and $h_p$. More details about the network architecture are given in Appendix A.

**Training procedure.** We now address the question of how to train the TLM. Focusing on fine-tuning scenarios, we finetune the original LLM $M_p$ and learn the semi-autoregressive model $M_q$ by maximizing the following objective:

$$\mathcal{L}(p, q) = (1 - \lambda)\mathbb{E}\Big[\sum_{i=1}^{T} \log p(w_i|w_{<i})\Big] + \lambda\mathbb{E}\Big[\frac{1}{k}\sum_{i=1}^{T-k}\sum_{j=1}^{k} \log q(w_{i+j}|w_{<(i+j)})\Big], \quad (8)$$

$$q(w_{i+j}|w_{<(i+j)}) = q_0(w_{i+j}|h_p(w_1, ..., w_i), h_q(w_{i+1}, ..., w_{i+j}))$$

The expectation is taken over the fine-tuned dataset. $T$ is the number of tokens in a sample from the dataset and $\lambda \in (0, 1)$ is a factor that balances the learning of $p$ and $q$. Similar to existing literature (Cai et al., 2024), we consider two training setups:

---

[2]When implemented as an RNN, the design is similar to the model proposed in Zhang et al. (2024a).

- *Separate training*. In this mode, we train $p$ and $q$ in two stages, where we set $\lambda = 0$ and $\lambda = 1$ in the respective stages. In each stage, we freeze the parameters of $p$ when training the $q$, and vice versa. This guarantees that the learning of $q$ will not affect $p$;

- *Joint training*. This corresponds to the case where we train $p$ and $q$ in one stage using a single $\lambda$. This mode allows us to learn a $p$ whose representation is also useful for predicting more future tokens, which can potentially lead to better draft quality. However, the learning of $q$ may also 'drag' that of $p$, leading to a potential degradation of $p$'s quality.

We use Q-LoRA (Dettmers et al., 2024) to train $p$ and $q$, where we assign different ranks to the parameters in $p$ and $q$: (a) For the parameters $\theta_p$ in $p$, which include both the network $h_p(\cdot)$ and the original LM head, we use a LoRA rank $r_p$ that is much smaller than the matrix size (e.g. $r_p = 32$); (b) For the parameters $\theta_q$ in the TLM $q$, which include both the network $h_q(\cdot)$ and the new LM head $q_0$, we use a LoRA rank $r_q$ that is relatively larger (e.g. $r_q = 4096$). See Figure 3 for more details.

## 5 RELATED WORKS

**Efficient inference schemes in LLMs** (Bai et al., 2024; Xu et al., 2024; Zhou et al., 2024) have received significant attention recently due to the ever-increasing size of such models. Approaches to improving efficiency range from static methods, such as pruning (Men et al., 2024) and quantization (Ashkboos et al., 2024), which reduce overall model size and hence computational requirements, to dynamic methods, such as early-exiting (Schuster et al., 2022; Bae et al., 2023) and hybrid models (Kag et al., 2022; Ding et al., 2022; 2024), which accelerate inference by adjusting the amount of computation based on the (estimated) difficulty of a given token or prompt. Our work is complementary to these general efficiency efforts (e.g., our semi-autoregressive scheme can be easily applied on top a quantized model). Note, however, that unlike our work and other speculative decoding frameworks, such classic approaches cannot guarantee that LLM's outputs remain identical.

**Non-autoregressive decoding** (NAR) accelerates inference by eliminating or relaxing the sequential dependencies between tokens (Gu et al., 2018). However, NAR often suffers from reduced accuracy compared to its autoregressive counterparts. Current efforts to improve performance focus on reintroducing some degree of conditional dependence between tokens, for instance, through generative flows (Ma et al., 2019) and conditional random fields (Sun et al., 2019). Additional refinements include iterative decoding strategies (Lee et al., 2018; Ghazvininejad et al., 2019), as well as improvements to training data and loss functions (Ding et al., 2021; Du et al., 2021).

**Speculative Decoding** (Leviathan et al., 2023) is a widely used framework for accelerating LLM inference. It employs a (smaller) draft model to propose multiple tokens at once, which are then verified by a (larger) target model. Initially, a separate transformer-based language model was used as the draft model (Leviathan et al., 2023; Chen et al., 2023; Kim et al., 2024). More recent work has shifted toward using (part of) the target model itself for drafting (i.e., self-speculative decoding). For example, Medusa Decoding (Cai et al., 2024; Gloeckle et al., 2024) combines non-autoregressive (NAR) techniques with speculative decoding by training multiple LM heads conditioned on the target model's final layer to generate draft tokens in parallel. Techniques like LayerSkip (Elhoushi et al., 2024) and FREE (Bae et al., 2023) integrate early-exit strategies by drafting with earlier layers and verifying with later layers, while Zhang et al. (2024b) adaptively skips intermediate layers for drafting. Recent works have explored the use of more light-weighted model as draft models, such as a n-gram model (Fu et al., 2024) and a small RNN (Zhang et al., 2024a), as discussed below.

**Light-weighted draft model design**. Like our work, concurrent works have also explored taking the draft model as a small autoregressive model conditioned on the hidden states of the original LLM (Ankner et al., 2024; Zhang et al., 2024a; Li et al., 2024; Nair et al., 2024). Among these works, Zhang et al. (2024a) is similar to one of the implementations in our design. Apart from differences in practical implementation details, the major differences to these works are (a) unlike these works which focus on specific implementations, our works focuses on high-level design, which naturally connects different implementations. The disentanglement of specific implementations and high-level design allows us to explore different architecture choices with various trade-offs in drafting efficiency and draft quality; (b) Unlike existing works which primarily focus on learning the draft model on pre-trained dataset e.g. ShareGPT (2023), our work focus on fine-tuning scenarios, where the draft model is trained on specific domain data jointly or separately with the original LLM.

|  | SQL-context | SAMSUM | GSM8K | ChatDoctor |
|---|---|---|---|---|
| *Domain* | code generation | text summarization | math reasoning | medical QA |
| *# examples* | $\sim$90k | $\sim$10k | $\sim$8k | $\sim$50k |
| *# draft tokens* | 4 | 3 | 3 | 4 |
| *LLM considered* | Phi-3 | LLama2-13B | Mistral-7B | Mistral-7B |

Table 1: Summary of the tasks considered. A smaller subset of ChatDoctor is used in experiments.

## 6 EXPERIMENTS

**Baselines**. We compare the proposed semi-autoregressive method (denoted as 'semi' henceforth) with the following two representative methods in (self-)speculative decoding, both of which can be seen as different implementations of the framework specified in Eq. 2.

- *Block autoregressive decoding* (block). Widely known as Medusa decoding (Cai et al., 2024), this corresponds to the case where we implement the draft model as Eq. 3, which ignores the dependence between the draft tokens.
- *Skip-layer decoding* (skip). This corresponds to the case where we implement the draft model as a single transformer model with fewer layers (Elhoushi et al., 2024), as in Eq. 4. Here we use 8 layers, following the setups in Elhoushi et al. (2024).
- *Recurrent drafter* (redraft). This corresponds to the design in (Zhang et al., 2024a), which can be seen as implementing $h_q$ in the proposed semi-autoregressive draft model (eq.5) as a RNN.

In addition to the above comparison, we also compare different implementations of the network $h_q$ in the proposed semi-autoregressive draft model. Some of these implementations are closely related to state-of-the-art methods such as Hydra (Ankner et al., 2024) and EAGLE (Li et al., 2024).

**Evaluation metrics**. We compare different (self-)speculative methods from the following angles:

- *Acceleration*. This metric is defined as the ratio between the wall time $t$ when decoding a sentence without speculative decoding and the wall time $t'$ when decoding with speculative decoding:

$$\texttt{Acceleration} := \frac{t}{t'}$$

- *Token acceptance rate*. This metric measures how many tokens drafted by the draft model $M_q$ are accepted by the original LLM $M_p$, which directly reflects the quality of the draft model $M_q$.
- *Extra memory cost*. This metric is defined as the ratio between the number of additional parameters $\theta'$ introduced in a specific design of $q$ and the number of parameters $\theta$ in the original LLM:

$$\texttt{MemoryCost} := \frac{|\theta'|}{|\theta|} \times 100\%$$

- *Generation quality of the target model*. Finally, when we compare joint learning and separate learning, we also investigate how different designs of the draft model $M_q$ will affect the generation quality of the original model. Theoretically, the draft model should have no impact on the original model $M_p$ in speculative decoding. However, when $M_p$ and $M_q$ are trained jointly, the training of $M_q$ may have an impact on $p$, as discussed in §4.2. Here, we measure generation quality by the Rouge-L score between models' generation and the ground truth.

**Tasks and models**. A summary of the tasks and models considered is given in Table 1. Specifically, we use the following datasets: SQL-context (b-mc2, 2023) for SQL code generation based on user queries, SAMSUM (Gliwa et al., 2019) for text summarization, GSM8k that include 8k grade school math questions and answers, and ChatDoctor (Yunxiang et al., 2023) which is a dialogue dataset for conversations between a doctor and a patient. For ChatDoctor, we use a subset of 50k in experiments.

**Computing resource**. Results on SQL-context, SAMSUM, and GSM8K are computed using two A10 GPUs in under a day. Results on ChatDoctor are computed with a single A100 GPU in a day.

|  | skip | block | semi | redraft |
|---|---|---|---|---|
| acceleration (↑) | 1.94× | **3.76×** | 3.71× | 3.68× |
| token acc rate (↑) | 75.1% | 73.8% | 74.4% | 74.6% |
| +memory (↓) | 4.38% | 12.4% | 8.51% | 9.52% |

(a) **SQL-context**

|  | skip | block | semi | redraft |
|---|---|---|---|---|
| acceleration (↑) | 1.32× | 1.71× | 2.02× | **2.07×** |
| token acc rate (↑) | 45.1% | 43.6% | 51.2% | 52.3% |
| +memory (↓) | 2.29% | 4.45% | 3.09% | 3.55% |

(b) **SAMSUM**

|  | skip | block | semi | redraft |
|---|---|---|---|---|
| acceleration (↑) | 1.51× | 2.55× | **2.63×** | 2.57× |
| token acc rate (↑) | 57.8% | 63.6% | 65.4% | 64.6% |
| +memory (↓) | 1.33% | 5.12% | 2.21% | 2.44% |

(c) **GSM8K**

|  | skip | block | semi | redraft |
|---|---|---|---|---|
| acceleration (↑) | 1.61× | 1.78× | **2.53×** | 2.51× |
| token acc rate (↑) | 60.1% | 48.3% | 62.4% | 61.0% |
| +memory (↓) | 1.04% | 3.18% | 1.69% | 1.95% |

(d) **ChatDoctor**

Table 2: The performance of different (self-)speculative decoding algorithms. *Skip*: the skip-layer decoding method by (Elhoushi et al., 2024). *Block*: the block autoregressive decoding method by (Cai et al., 2024). This method is also known as Medusa. *Semi*: the proposed semi-autoregressive decoding method (implemented with a simplified transformer). *Redraft*: the method in (Zhang et al., 2024a) which implements $h_q$ as a RNN. Our method offers highest acceleration on the majority of the tasks while requiring a reasonable memory cost.

**Training setups**. We employ the same training procedure for all methods, as outlined in §4.2. For joint training, we set $\lambda$ in the objective function (Eq. 8) to 0.25. This ensures that the learning of $p$ predominates the training process, so as to guarantee the generation quality of the original LLM. Other details such as model architecture and optimizer settings can be found in Appendix B.

**Inference setups**. During inference, we enable KV caching but disable tree attention (Cai et al., 2024). We disable tree attention (Cai et al., 2024) to focus our evaluation on the quality of the draft model itself, though our method is fully compatible with tree attention.

**Number of draft tokens**. We look ahead 3 - 4 tokens for all speculative decoding methods considered, depending on the task. Looking further ahead doesn't result in any acceleration improvement.

When presenting the results, we report the results from the simplified transformer design for $h_q$ (see Eq. 6) in the main table. The LSTM design described in Eq. 7 achieves a similar performance and a comparison between the two implementations is provided in Figure 4.

## 6.1 MAIN RESULTS

**Comparison with other decoding methods**. Table 2 compares the proposed semi-autoregressive decoding method with two baseline approaches. Overall, our method consistently achieves the highest acceleration in three out of the four tasks considered, with the exception of the SQL-context dataset, where block decoding marginally outperforms our method. We hypothesize that this is due to the nature of SQL code, where the fixed syntax allows block decoding to be equally effective by ignoring the draft token dependencies. We expand on this further in § 6.2.

The superior acceleration observed in the other tasks can be attributed to two factors: (i) the high draft quality of our method, as reflected by its higher acceptance rate comparable to or even exceeding that of a pruned transformer (as seen in skip-layer decoding); and (ii) its low drafting latency, which is negligible compared to even a single decoding layer in a transformer. A more detailed analysis of execution time is presented in Fig. 4a. These results demonstrate that our method strikes a better balance between draft quality and drafting latency.

In terms of memory cost, our method has an advantage over the block decoding method, as it eliminates the need to maintain multiple language model (LM) heads for the draft model. However, our method is less memory efficient than skip-layer decoding, as it requires additional memory to store the TLM for processing the draft tokens $d_i$. This introduces a minor memory overhead.

**Comparison among different implementation of** $h_q$. To gain further insight into the trade-off between draft quality (i.e. the acceptance rate) and inference latency of the draft model, we compare different implementations of $h_q$ with varying complexity, as shown in Fig. 4. We specifically compare to the case where $h_q$ is realized as a standard transformer with varying number of layers. Al-

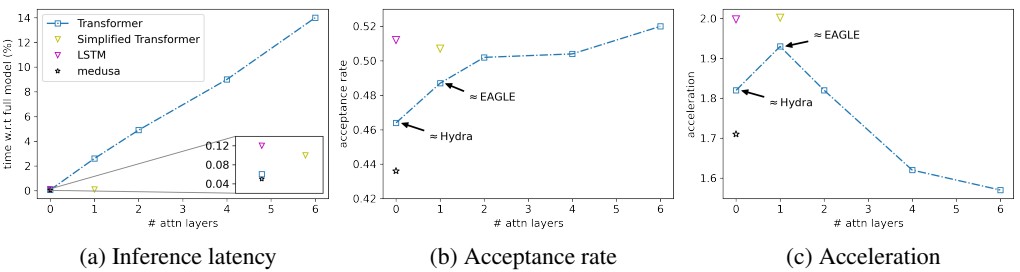

(a) Inference latency        (b) Acceptance rate        (c) Acceleration

Figure 4: Comparison of different implementations of $h_q(\cdot)$ on SAMSUM, including Medusa, simplified transformer, LSTM, and standard transformer (denoted as Transformer) with varying number of layers. In the standard transformer implementation, the zero-layer case corresponds to taking the embedding $e(d_j)$ of the last token $d_j$ as the transformer's output, which is similar to Hydra (Ankner et al., 2024); see Appendix C for further details.

though not identical, this implementation is closely related to two existing methods Hydra (Ankner et al., 2024) and EAGLE (Li et al., 2024); see Appendix C for a discussion. These results are collected from the first 200 samples of the test set in the SAMSUM dataset.

The results clearly indicate that implementing $h_q$ as either a simplified transformer or an LSTM achieves a good trade-off between the two metrics: the inference latency in these two models is negligible even compared to a *single* transformer layer (see Fig. 4a), yet offering an acceptance rate comparable to a 6-layers transformer (see Fig. 4b). This leads to considerable acceleration as shown in Fig. 4c. This result verifies our hypothesis that even simple models are competent in handling a small number of draft tokens. On the other hand, although increasing the number of attention layers in a transformer does improve the acceptance rate owing to the increased capacity, it fails to deliver satisfactory acceleration due to higher inference latency. These results confirm our hypothesis that using standard transformer layers might be unnecessary to process a small number of (draft) tokens, and simpler designs like ours are sufficient.

Another interesting result emerges when comparing our method with a transformer that has zero attention layers, which directly takes the embedding $e(d_j)$ of the last draft token as the transformer's output, i.e. $h_q(\mathbf{d}_{\leq j}) = e(d_j)$. Unlike our approach, this setup only considers the last draft token $d_j$ while ignoring all previous draft token $\mathbf{d}_{<j}$ when predicting $d_{j+1}$, resulting in a comparatively lower acceptance rate. This finding highlights the importance of considering the entire draft token sequence $d_1, ... d_j$ rather than solely considering $d_j$ for predicting $d_{j+1}$. It also justifies our design of a lightweight LSTM or a simplified transformer to process the draft tokens, which is able to handle draft sequence of varying lengths while remaining computationally efficient.

## 6.2 FURTHER ANALYSIS

**Structured vs. unstructured data**. From the results in Table 2, we find that the acceleration gap between our method and the block decoding method is small on code generation and math reasoning tasks. This could be due to: (i) model differences; see Table 1; and (ii) data differences, with the data in these tasks being more structured compared to others. For example, in the SQL generation task, the decoding space is smaller, making the additional dependencies introduced by our method less impactful[3].

To further investigate this, we conduct a more extensive comparison between our method and the block decoding method on tasks with highly structured data, where we eliminate the impact of model differences. The results in Table 3 indicate that our method only marginally improves acceleration on these tasks for both Mistral-7B and Phi-3. Specifically, while our method consistently achieves a higher acceptance rate for both models, the improvement is marginal ($\leq 5\%$), suggesting that there are no substantial benefits for additionally considering the dependencies between the draft tokens in tasks with constrained syntax, which results in a marginal gain in acceleration.

---

[3]For example, in SQL generation task, given the token 'SELECT' at the first position, the tokens 'FROM' and 'TABLE' are likely to appear in subsequent positions, so the token 'TABLE' does not depend on 'FROM'.

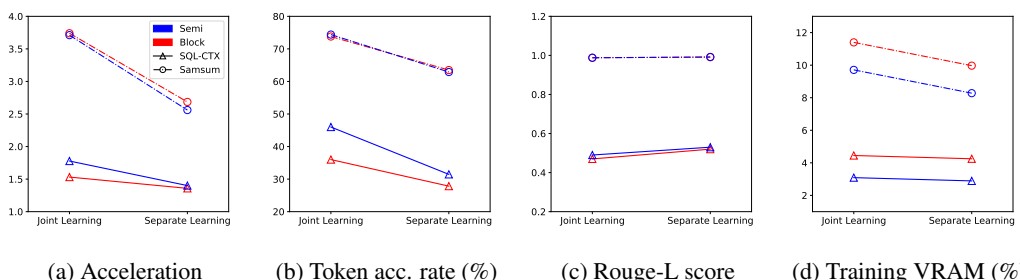

(a) Acceleration (b) Token acc. rate (%) (c) Rouge-L score (d) Training VRAM (%)

Figure 5: Comparison between joint and separate training. Joint training typically offers a better acceptance rate and hence better acceleration, but may incur a slight drop in accuracy.

|  | acceleration | | token acc rate | | | acceleration | | token acc rate | |
|---|---|---|---|---|---|---|---|---|---|
|  | block | semi | block | semi |  | block | semi | block | semi |
| Mistral | $3.44\times$ | $3.39\times$ | 69.2% | 70.4% | Mistral | $2.55\times$ | $2.63\times$ | 63.6% | 65.4% |
| Phi-3 | $3.76\times$ | $3.71\times$ | 73.8% | 74.4% | Phi-3 | $1.74\times$ | $1.78\times$ | 58.2% | 63.2% |

(a) **SQL-context** (b) **GSM8K**

Table 3: A more detailed comparison with the block method on tasks with highly structured data.

**Joint vs. separate training**. We conducted experiments to compare these two training modes on the SAMSUM and SQL-context datasets. The key questions are: (a) how does joint training affect final acceleration, and (b) to what extent does joint training affect generation quality?

Fig. 5 presents a comparison of joint and separate training from four perspectives. Overall, joint training consistently achieves a significantly higher token acceptance rate and faster acceleration (Fig. 5a and Fig. 5b), particularly for the SQL generation task. This improvement can be attributed to better coordination $p$ and $q$ in joint training, where the model $M_p$ learns a representation that not only predicts the next token but also aids in predicting subsequent tokens. At the same time, we see that joint training does have a small impact on generation quality, as measured by the Rouge-L score (Fig. 5c). This minimal quality loss may be explained by our use of a relatively small factor $\lambda$ in the objective function (Eq. 8), where the training of the main model $M_p$ dominates the learning process.

Fig. 5d further compares the VRAM usage during training. The VRAM usage corresponds to the number of parameters in the LoRA weights. The VRAM usage of joint training mode is slightly higher than that of the separate training mode, as the former needs to simultaneously store the LoRA weights of the target model $M_p$ and that of the draft model $M_q$.

Based on the above results, we recommend to use joint training whenever possible, but with careful attention to any potential drop in accuracy. However, we found this drop to be negligible in practice.

# 7 CONCLUSION

In this work, we proposed a unified probabilistic framework for token dependency modeling and classified existing literature based on how they attempted to model the *recent* and *distant* token dependencies. We then analyzed the trade-off regarding drafting speed and acceptance rate explored in prior works. Based on these insights, we proposed an improved semi-autoregressive draft model, that processes the *distant* and *recent* tokens by the original LLM and TLM respectively for fast drafting while retaining a high acceptance rate. We proposed two variants of our scheme based on (i) a simplified transformer, and (ii) an LSTM. We validated the design of our draft model via experiments on four distinct applications, realized via model finetuning, where our model outperformed other competing methods in 3 out of 4 settings, and performed on par in the structured prediction task where the modeling of recent token dependencies carry negligible value. We further analyzed the reasons for these improvements, and highlighted several interesting avenues for future work.

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

## A  FURTHER IMPLEMENTATION DETAILS

**Tree attention**. Tree attention is a technique originally used in multi-tokens generation to further accelerate inference. It works by considering multiple candidate tokens concurrently rather than focusing solely on the most likely candidate. Similar to prior methods (Cai et al., 2024; Luk et al., 2024; Ankner et al., 2024; Li et al., 2024), our approach is fully compatible with tree attention due to the negligible computational cost of the draft model.

**KV Caching**. Key-Value (KV) caching is a crucial technique for optimizing the efficiency of attention mechanisms by avoiding the recompution of previous KV pairs when generating subsequent tokens. In our approach, the KV caches for the draft tokens $d_1, ..., d_K \sim q$ generated by the proposal model $q$ are absent in the draft LLM, posing challenges for continued generation. To address this, we simply recompute the missed caches during the verification stage of speculative decoding. This process can be done in parallel for all draft tokens, which remains highly efficient[4].

**Network architectures**. We provide details about the LSTM and the MLP used in the proposed semi-autoregressive draft model. Below, we use $H$ and $V$ to denote the size of the hidden states of the LLM and the vocabulary size respectively. Note that this size is the same as the size of the token's embedding.

- *MLP in the simplified transformer*. This MLP has two layers, each of which has $H/2$ neurons. The activation function in the MLP is the same as the activation function in the LLM. Therefore the output of the simplified transformer $h_q(d_1, ..., d_K) \in \mathbb{R}^{H/2}$;

- *LSTM*. The LSTM has two layers, where the hidden states are of size $H/2$. We use tanh as the activation function in the LSTM. We take the hidden states corresponding to the last token as the output of the LSTM. Therefore the output of the LSTM $h_q(d_1, ..., d_K) \in \mathbb{R}^{H/2}$;

- *MLP in the LM head*. Recall that this MLP takes both $h_p$ and $h_q$ as inputs to predict the next draft token. The MLP $f$ computes the output as $f(h_p, h_q) = W_3^\top \sigma(\text{concat}(W_p^\top h_p, W_q^\top h_q) + b_3,$ where the weight matrices $W_p \in \mathbb{R}^{H \times 1024}$ and $W_q \in \mathbb{R}^{H/2 \times 1024}$. The matrix $W_3 \in \mathbb{R}^{2048 \times V}$. Here $\sigma(\cdot)$ is the activation function, which is the same as the activation function in the LLM.

## B  FURTHER EXPERIMENT DETAILS

|  | SAMSUM | SQL-context | GSM8K | ChatDoctor |
|---|---|---|---|---|
| *LoRA rank of p* | 8 | 32 | 16 | 32 |
| *LoRA rank of q* | $0.75H$ | $0.75H$ | $0.35H$ | $0.5H$ |
| *# training epochs* | 4 | 2 | 2 | 2 |
| *effective bs* | 4 | 8 | 4 | 16 |
| *Optimizer used* | adamw-fused | adamw-fused | adamw-fused | adamw-fused |
| *learning rate* | 1e-4 | 2e-4 | 1e-4 | 2e-4 |
| *GPU* | $2 \times$ A10 | $2 \times$ A10 | $2 \times$ A10 | $1 \times$ A100 |

Table 4: Summary of the detailed training setups.

---

[4]Under the setup of KV caching, speculative decoding does not reduce the overall number of arithmetic operations, as the KV caches corresponding to the draft tokens must still be recomputed eventually. However, the memory bandwidth of the GPU is significantly improved, as it can now process multiple draft tokens simultaneously rather than handling them sequentially. This is akin to the principles of Flash Attention.

# C    CONNECTION TO RELATED METHODS

In section 6.1, we compare several implementations of the network $h_q$ for processing draft tokens $d_1, ...d_j$ in the proposed semi-autoregressive draft model. One implementation we considered is a transformer with varying number of layers. We discuss here how this implementation is related to wo state-of-the-art draft model designs: Hydra (Ankner et al., 2024) and EAGLE (Li et al., 2024).

**Connection to Hydra** (Ankner et al., 2024). Given previously accepted tokens $\mathbf{w}_{\leq i}$ and previous draft tokens $\mathbf{d}_{\leq j} = \{d_1, ...d_j\}$, Hydra predicts the next draft token by jointly sending the embeddings of all current draft tokens $\{e(d_1), ...e(d_j)\}$ and the hidden states $h_p(\mathbf{w}_{\leq i})$ from the original LLM to a LM head (with a total of $K$ LM heads). The mentioned implementation of $h_q$ as a standard transformer with zero decoding layers can be viewed as a simplified version of Hydra, where the model only uses the embedding of the most recent draft token $e(d_j)$ and the hidden states $h_p(\mathbf{w}_{\leq i})$ from the original LLM in prediction, ignoring $e(d_1), ...e(d_{j-1})$. This setup, while being slightly less flexible due to the lost in draft token dependence, is more memory efficient as it only requires storing one LM head instead of $K$ different LM heads.

**Connection to EAGLE** (Li et al., 2024). Given previously accepted tokens $\mathbf{w}_{\leq i}$ and previous draft tokens $\mathbf{d}_{\leq j} = \{d_1, ...d_j\}$, EAGLE predict the next draft token $d_{j+1}$ by using a transformer decoder layer which takes inputs as both the hidden states of the current sentence and also the draft tokens. The output of this decoding layer, combined with the original hidden states, is then passed to an LM head to predict the next draft token. When we implement $h_q$ as a standard transformer with *one* decoding layer, it can be seen as a simplified and more streamlined version of EAGLE, where attention is only applied among the draft tokens.

