# OpenReview forum: "Semi-autoregressive Decoding for Efficient LLM Inference"
_ICLR.cc/2025/Conference — Submitted to ICLR 2025_

### Official Review · Reviewer_6TCM · 2024-10-28

**Soundness:** 2
**Presentation:** 4
**Contribution:** 2
**Rating:** 3
**Confidence:** 4

**Summary:**

This paper presents a method for efficient inference in LLMs, based around speculative decoding where a cheap drafter model/algorithm is used to generate a sequence of outputs that are then validated in a larger model. This paper specifically contributes a cheap mechanism (based on semi-autoregressive generation) for generating fast but high quality draft sequences, and show that this can improve overall runtime versus several baseline methods.

**Strengths:**

This paper is very well written, and easy to follow. I had no trouble understanding the motivation, algorithm, or experimental evaluation, and the design decisions underlying the technique were convincingly argued. The exploration of simple architectures for the drafter is interesting, and includes some very light models with few parameters.

**Weaknesses:**

I have three issues: novelty, need for joint training, and quality of experimental setup.

1. On novelty, a key paper is not cited:
Tandem Transformers for Inference Efficient LLMs; Aishwarya P S, et al 2024 https://arxiv.org/abs/2402.08644
This work performs block-generation from the drafter model with lagged cross-attention into the larger model, and includes experimentation with SPEED. The way in which the models are combined may be different, but the idea appears to be the same. At the very least your approaches should be compared analytically, or - better - empirically.

2. It is not clear why there is a need to finetune the original LLM. The fact this is shown to help in the experiments suggests to me that the data signal in the loss function (8) is quite different from the original LLM's output. That is, there's a problem of domain shift such that when the drafted is tuned with lambda=1 there ends up being a disconnect between that and the original LLM. An easier way to fixing this than tuning the LLM is to reframe the loss as a distillation objective, such that you train the drafter against the outputs of a frozen teacher. This is much cleaner than fine-tuning the original LLM to several different domains.

3. Experiments use a block size of 3-4 draft tokens. Compared to prior work on speed, e.g., the Tandem paper above,  Elhoushi et al, 2024, Medusa (last two of these cited in the submission), experiment with much bigger blocks, e.g., 20+ tokens. Maybe there are benefits to be had from using a small block (and a simpler model), however this disconnect is jarring, leading me to distrust the results in the paper. Line 406 says looking further ahead doesn't help, but perhaps this is because the drafter has been oversimplified.

Another jarring issue is the choice of datasets and models, namely that there is no overlap I can see with the prior work. I suggest adding datasets to facilitate easier comparison. Why did the model change between most evaluation tasks (Table 1)? Sticking to one model for the primary results would remove one confound, then report an ablation of the method across models separately.

**Questions:**

282: It's not clear how the original model vectors and drafter model vectors are used together. From the graphic is seems they are concatenated and then treated the same (weighted with a LORA matrix). I'd also appreciate more motivation for the asymmetry in LORA size.

Figure 4: it wasn't clear what the x-scale was on the zoomed box in 4a.

---

> ### Author Response · Authors · 2024-11-25
> **Clarification on weaknesses and questions + citation added for Tandem Transformer**
>
> We thank the reviewer for their insightful feedback and constructive suggestions. We address each concern as follows:
>
> **Response to Weaknesses**
>
> **1. Citation of Tandem Transformers**
>
> We appreciate the mention of the *Tandem Transformers* paper whose high-level idea aligns with the semi-autoregressive framework we present. We have cited this work and acknowledged its relevance in revision.
>
> While both our work and Tandem Transformer uses a small size transformer as the small model, the transformer in our method is even more light-weighted, which only contains 1 layer and fixed attention weights. Furthermore, our primary focus is on fine-tuning setup, contrasting with the pretraining set up of the Tandem paper. We have included this analytical comparison to strengthen our discussion in revision.
>
> **2. Why there is a need to finetune the original LLM**
>
> We apologize for the confusion made. **In short, this is due to our focus on fine-tuning scenarios (i.e. training a LLM to adapt to a downstream task)**, as opposed to pre-training scenarios where the LLM is kept fixed.
>
> Specifically, the problem we consider here is ‘how to accelerate inference for a fine-tuned LLM’, as opposed to 'how to accelerate inference for a fixed LLM'. Here, in addition to standard fine-tuning routine which solely train the LLM with e.g. LoRA, we also train an additional domain-specific module (the small language model) to assist inference. The training of this additional module can either be done jointly during the fine-tuning of LLM, or can be done after the fine-tuning.
>
> We appreciate the reviewer’s suggestion on KD, which is a cleaner and effective way to align the draft model with the LLM. We will try this objective in subsequent refinement.
>
> **3. Draft token size**:
>
> We argue that the optimal draft token size reported in previous work is **within similar ranges to** ours. For example, Medusa uses a draft token number of 3-4 (This is reported in Figure 7 in the appendix of the Medusa paper.) Elhoushi et al. 2024 uses a draft token size of 4/6/12 depending on the downstream task. However, longer draft token size (12) doesn’t imply improvement on speed gains in their experiment either (e.g. only 1.8x acceleration with 12 tokens).  Similarly, the *Tandem Transformers* paper reports results primarily with a block size of 7.  Our choice of optimal draft token size aligns with these prior studies findings. We also would like to emphasise that larger draft token sizes doesn’t guarantee more speed-up.
>
> **4. Dataset and model choices**
>
> We acknowledge that the dataset used in our work differs from prior studies like Medusa and Tandem Transformers. This is because **our focus is on fine-tuning scenarios**, where the LLM is trained on fine-tuning datasets, unlike prior works that rely on pre-training datasets. In our study, four commonly used fine-tuning datasets was selected to thoroughly compare different draft model designs in fine-tuning settings.
>
> For model selection, we evaluated diverse open-source models (Phi, LLaMA, Mistral) across these datasets to highlight the wide applicability of our approach. While reporting results across all models and datasets as the reviewer suggested would be ideal, resource constraints required us to selectively apply models to datasets. Importantly, we have controlled variables whenever possible—for example, using *Mistral* on both GSM8K and ChatDoctor to isolate dataset effects, and other models to minimize biases due to model differences.
>
> **Response to Questions**
>
> - **Clarification for the asymmetry in LORA size.**  We apologize for this confusion. As discussed earlier, the focus of our work is on fine-tuning scenarios, where we simultaneously fine-tune the original LLM and train a new module to accelerate inference. Since the original LLM is already pre-trained, it requires only a small LoRA size to adapt to the fine-tuning task. On the other hand, the new module (the small autoregressive model) is trained from scratch, and therefore requires a larger LoRA size to capture the necessary representations effectively. This asymmetry in LoRA sizes can be seen as a strategic decision to allocate LoRA ranks efficiently under resource-limited settings.
> - **Clarification on Figure 4:**  Thank you for noting the ambiguity in Figure 4a. We will revise the figure to indicate that the three symbols on the left represent 0 attention layer, and the green triangle represents 1 attention layer.

---

> > ### Comment · Reviewer_6TCM · 2024-11-26
> >
> > I thank the authors for the clarifications. My main issue still stands. I don't see understand the focus on fine-tuning while drafting, which doesn't appear well motivated. The approach taken by prior work is to learn a drafter to improve the inference efficiency of a fixed model over any domain of input or task. This is cleaner, as efficient drafting and model adaptation are two different problems, that can be handled separately. A stronger argument -- with empirical justification -- is needed for treating these jointly.

---

> ### Author Response · Authors · 2024-11-28
> **Clarifying the Motivation for Jointly Considering Fine-Tuning and Drafter Training**
>
> We deeply appreciate your insightful comment and prompt response. We completely agree that the approach taken by prior work is cleaner. That said, we believe there are still some important practical reasons for jointly considering fine-tuning and draft learning, as detailed below.
>
> - **Mismatch between fine-tuned LLM and pre-trained drafter**. After fine-tuning a LLM, the pre-trained drafter may become outdated, especially when there is a significant distribution shift in the fine-tuned data (for example, consider a private dataset in finance). As highlighted in [r1], the gap between the acceptance rate of a static drafter and a domain-specific drafter can be as large as 30% or more. This serves as a strong motivation for simultaneously fine-tuning the LLM and the drafter;
> - **Practices in modern LLM serving**. In many industrial LLM systems, requests are typically not handled by a single centralized pre-trained model. Instead, they are routed to multiple domain-adapted models in the backend to ensure optimal response quality (via a request router similar to [r2]). To accelerate these domain-specific LLMs, jointly training a domain-specific drafter that aligns well with the LLM seems to be a natural choice;
> - **Expanding the applicability of speculative decoding methods**. Not all researchers or practitioners have the resources to train a drafter on a pre-trained dataset. Our work on successfully applying speculative decoding in fine-tuning scenarios makes this technique  more accessible to users with limited resource (e.g. 1~2 A10 GPUs).
>
> In summary, the joint consideration of fine-tuning and efficient drafting has important implications for both *industry* and *academia*. We hope these explanation address your concerns. At the same time, we would be eager to hear any further thoughts or insights you may have on this. Thank you!
>
>
> *References*
>
> [r1] liu et. al. Online Speculative Decoding. ICML 2024.
>
> [r2] Ong et.al. RouteLLM: Learning to Route LLMs with Preference Data. arxiv 2406.18665.

---

### Official Review · Reviewer_M39m · 2024-10-31

**Soundness:** 2
**Presentation:** 3
**Contribution:** 3
**Rating:** 5
**Confidence:** 3

**Summary:**

This paper proposes a semi-autoregressive decoding method aimed at accelerating LLM inference. Compared with two existing approach: 1) block autoregressive model and 2) pruned autoregressive model, this work introduces a new framework that combines the advantages of both: the improved latency of the block autoregressive model and the high acceptance rate of pruned autoregressive models. Specifically, in the proposed framework, preceding tokens are encoded using the large target model, while draft tokens are encoded with a smaller model, such as a one-layer transformer or RNN. To train the smaller model, the authors experiment with two training strategies: separate training and joint training with Q-LoRA. The experiments demonstrate that the proposed semi-autoregressive decoding approach achieves promising results in both acceptance rate and inference latency.

**Strengths:**

1.	This work unifies and analyzes existing speculative decoding methods, proposing a semi-autoregressive framework that effectively balances acceptance rate and low latency.
2.	The paper is well-organized and clearly written.

**Weaknesses:**

1.	The novelty of this paper is limited.

2.	This paper lacks direct comparisons with previous works under the same settings. For example, in the GSM8K code generation dataset, Eagle achieves a 3.x speedup, while this paper achieves only 2.6. More discussion on this is needed.

**Questions:**

1. Have you compared your methods with Medusa or Eagle under the same settings? It appears that all the results listed are from your experiments.

2. The use of distance token representations in the tiny model is unclear. In the Appendix, the MLP in the MLP head takes the concatenation of h_p and h_q as input. However, h_q maybe the size of SH, where S in the distance token number. Did you use the last token representation or some other methods? Did I miss some details?

---

> ### Author Response · Authors · 2024-11-25
> **Novelty of the paper and comparisons with existing work**
>
> **Response to Weaknesses**
>
> **1. On novelty of the paper**
>
> We appreciate the reviewer’s critical feedback and acknowledge the existence of similar paradigms, such as Medusa and EAGLE. However, we believe our work introduces significant distinctions and contributions in the following key aspects:
>
> - *Focus on fine-tuning scenarios.* While most prior studies such as Medusa and EAGLE have focused on pre-training settings where the draft model is trained on pre-training datasets like ShareGPT, the application of these techniques in fine-tuning scenarios remains underexplored. Our research addresses this gap by investigating not only the optimal draft model design but also the most effective training strategies across four fine-tuning tasks.
>
> - *Unified framework for trade-off analysis.* Existing works propose different draft model designs but treat them largely as orthogonal approaches, without a comprehensive study of the underlying trade-offs. We bridge this gap by introducing a unified framework for draft model design, systematically analyzing the balance between draft quality and drafting speed.
>
> - *New draft model design*. Finally, we propose a novel semi-autoregressive draft model, which handles the dependencies over recent and distant tokens by a large model and a small model respectively. This new way of dependencies modeling allows us to use any off-the-shelf small language model with diverse architectures and complexities, as opposed to existing designs which often relies on a specific architectural choice. This flexibility expands the design space, enabling the discovery of new implementations with improved performance.
>
> **2. On comparisons with existing work and why the result reported in the EAGLE paper is higher**
>
> We acknowledge that our evaluation setup does not entirely align with those of Medusa and Eagle. However, this difference arises from our specific focus on application and evaluation. Specifically:
>
> - *Difference in applications*. Our work primarily targets fine-tuning scenarios, whereas prior works like Eagle and Medusa learn their draft model on pre-trained datasets such as ShareGPT. Consequently, the draft model in our experiments is trained on less data, making it inherently less powerful than those used in Eagle or Medusa. This leads to smaller speedups in our results.
>
> - *Difference in evaluation focus.*  The original work of Medusa / EAGLE study the acceleration brought by **the combined use of draft model and tree attention**, thus leading to higher acceleration. In our evaluation, we focus on comparing different draft model designs solely by disabling tree attention, aiming to understand the trade-off between acceptance rate and drafting latency in different designs. This choice also ensures a fairer comparison with other baselines, such as SkipLayer, which cannot leverage tree attention. Having said that, our method can be further accelerated when used in combination with tree attention.
>
> We hope the reviewer finds this difference in application scope and evaluation focus understandable. In our revision, we will further clarify these distinctions to avoid potential confusion.
>
> **Response to Questions**
>
> **Q1**: Thanks for the question. Please see our response to your weakness 2 above.
>
> **Q2**: We sincerely apologize for the confusion made. Yes, the reviewer’s understanding is indeed correct.
>
> In our design, h_q is a tiny language model (e.g. a LSTM or a simplified transformer) used to computed the representation of the sequence of recent tokens d = {d_1, d_2, … d_K}. We take the hidden states corresponding to the last tokens d_K as the output of h_q, which serves as the summary of the sequence d = {d_1, d_2, … d_K}.
>
> We have revised the manuscript to clarify this explanation (please see 'network architecture' in the updated Appendix A). Please do not hesitate to let us know if this addresses your concern or if additional clarification is needed.

---

### Official Review · Reviewer_fPR5 · 2024-11-02

**Soundness:** 2
**Presentation:** 3
**Contribution:** 2
**Rating:** 5
**Confidence:** 4

**Summary:**

The paper presents a semi-autoregressive decoding approach aimed at enhancing the inference efficiency of large language models (LLMs). It offloads some computational burden from the original LLM to a smaller, more efficient autoregressive model. This approach separately handles long-term dependencies using the large model and short-term dependencies with the smaller model, striking a balance between draft quality and drafting speed. Experiments across text summarization, medical QA, code generation, and mathematical reasoning tasks demonstrate the efficacy of the proposed method in terms of acceleration, memory cost, and training convenience. The study contributes to the field by offering a practical solution for efficient LLM inference, which is crucial for real-world deployment of these models.

**Strengths:**

1. The paper introduces a semi-autoregressive decoding paradigm that attempts to optimize the balance between draft quality and inference speed in LLMs.
2. The proposed method demonstrates potential in accelerating inference across various text generation tasks.

**Weaknesses:**

1. The approach of integrating a small autoregressive model is not novel and has been previously explored in Cheng et al. (https://arxiv.org/abs/2403.09919) The paper does not sufficiently differentiate its method from existing works and misses comparison in experiments, which is a significant oversight.

2. The study's scope is confined to models no larger than 13 billion parameters. There is an absence of discussion and experimentation regarding how the proposed method scales and performs as model size increases. This is a critical gap, as the acceleration benefits and computational efficiency are likely to vary with larger models.

3. While the paper initiates a meaningful discussion on the types of tasks that could benefit from the proposed method, it does not delve deeply enough into this analysis. A more comprehensive investigation into which task types experience significant acceleration and the underlying reasons would have strengthened the study's conclusions and practical implications.

**Questions:**

1. The study is limited to models no larger than 13 billion parameters. Could you discuss how you anticipate your method would perform as model size increases, especially given the trend towards larger LLMs? Are there any scalability concerns or potential modifications to the approach that you foresee as necessary for larger models?

2. The paper includes a discussion on the types (Sec. 6.2. structured vs. unstructured data) of tasks that benefit from your proposed method but does not delve deeply into this analysis. Could you provide a more in-depth exploration of which task types are most likely to see significant acceleration with your method, and what factors contribute to these differences? This could include a discussion on the nature of the tasks, the complexity of the inputs,  the models' original ability on the tasks, and how these factors interact with the decoding process.

3. Table 2 shows that the skip method achieves a lower token acceptance rate compared to the proposed model, which contradicts intuition. Typically, one would expect the performance of the skip method to be closer to that of the original model. The manuscript does not provide a clear explanation or empirical evidence to support this claim, which undermines the credibility of the findings.

---

> ### Author Response · Authors · 2024-11-26
> **Rebuttal by the authors (1/2)**
>
> We appreciate your detailed feedback and the constructive criticism, which we found highly valuable in improving our work. In the text below, a detailed response to your concerns and questions are provided.
>
> **Weakness**
>
> **1. The approach of integrating a small autoregressive model is not novel and has been previously explored in Cheng et al**
>
> Thank you for the constructive critique. We acknowledge that one specific implementation of our proposed model is similar bears similarities to the work of Cheng et a. Following the reviewer’s valuable suggestion, we have now added a comparison to this work. The result shows that the two designs do achieve similar performance.
>
> That being said, we would like to emphasize two major differences between our work and Cheng et al:
>
> - *General model vs specific implementation*. A crucial difference between our work and Cheng et al is that they primarily focus on a specific implementation of small autoregressive model (RNN), whereas our work proposed a general framework of semi-autoregressive model. This broader perspective enables us to systematically explore various architectural choices for the small model, balancing trade-offs in efficiency and capacity. This exploration, as demonstrated in Figure 4 of our experiments, provides fine-grained insights that were not investigated in Cheng et al.'s work, justifying why the design in our work (and also the similar design in Cheng et al’s work) is optimal.
> - *Specific focus on fine-tuning.* Another significant difference is our focus on finetuning for downstream applications. While Cheng et al. primarily train their draft model on large pre-training datasets such as ShareGPT, we target fine-tuning scenarios—an underexplored area in speculative decoding research. Fine-tuning introduces unique challenges, including limited training data and constrained training resources, which require distinct strategies. These challenges were not addressed in Cheng et al.'s work.
>
> **2. The study's scope is confined to models no larger than 13 billion parameters. There is an absence of discussion and experimentation regarding how the proposed method scales and performs as model size increases.**
>
> Thank you for raising this valid concern. We fully acknowledge the importance of evaluating the scalability of our method across models of varying sizes. Experiments on larger models are already underway, but due to time and resource constraints, the results are not yet available.
>
> In our current study, we chose to prioritize evaluating the method on *different classes* of models (e.g., LLaMA2, Phi-3, Mixtral) rather than exploring models of varying sizes within the *same class* (e.g., Vicuna-7B, Vicuna-33B, Vicuna-70B) as done in Cheng et.al. This decision was made to confirm the wide applicability and robustness of our approach across a diverse range of model types. We believe this is equally valuable as studying scalability within a single class of models.
>
> On the other hand, based on our observations and prior literature, we anticipate that the acceleration achieved by our method may decrease as model size increases (please see our response to Q1 for a discussion).
>
> We appreciate the reviewer's insight and will include a discussion on this limitation and our rationale for the chosen experimental scope in the revised manuscript.
>
>
>
> **3. While the paper initiates a meaningful discussion on the types of tasks that could benefit from the proposed method, it does not delve deeply enough into this analysis**
>
> Thank you for the insightful suggestion! Let us provide more details on what type of dataset are more likely to benefit from our method. Specifically:
>
> - **Structured datasets (e.g., SQL-context):** In tasks like generating SQL queries from prompts such as "What is the greatest number of wins by Japanese Formula Three?", the next tokens (e.g., ‘SELECT MAX(wins)’) are deterministic given the prompt, with no alternative continuations. In such cases, multiple tokens can be predicted simultaneously without considering their dependencies, so our method offers little advantage over simpler approaches like Medusa.
>
> - **Unstructured datasets (e.g., SAMSUM):** For tasks like summarizing dialogue (e.g. ‘Amy: Hi Tom, are you busy tomorrow … ’), subsequent tokens heavily depend on previous tokens (e.g., whether the summary starts with ‘Amy’ or ‘Tom’ leads to entirely different continuations). Here, our method captures these fine-grained dependencies effectively, offering significant benefits over Medusa.
>
> This explanation and analysis will be included in the final version of the paper. We are keen to learn from the reviewer whether our analysis above has provided a solid explanation on the impact of task types.

---

> ### Author Response · Authors · 2024-11-26
> **Rebuttal by the authors (2/2)**
>
> **Questions**
>
> **1. The study is limited to models no larger than 13 billion parameters. Could you discuss how you anticipate your method would perform as model size increases**
>
> Thank you for this insightful question. We anticipate that *the acceleration achieved by our method will decrease as model size increases*. This is because the performance gap between the main model and the draft model tends to widen as the main model scales, leading to a drop in the token acceptance rate. This trend aligns with observations in prior studies that utilize lightweight draft models for speculative decoding, such as Medusa and Cheng et al.
>
> We appreciate the opportunity to address this scalability consideration and will include this discussion in the revised manuscript for greater clarity.
>
> **2. Could you provide a more in-depth exploration of which task types are most likely to see significant acceleration with your method, and what factors contribute to these differences?**
>
> Please see our responses to weakness 3
>
> **3. Table 2 shows that the skip method achieves a lower token acceptance rate compared to the proposed model, which contradicts intuition.**
>
> Thanks for raising this important question. While this result may seem counterintuitive, it is indeed reasonable and represents one of the key findings of our work.
>
> Specifically, the relatively low acceptance rate in the skip method compared to our model is due  to two main factors:
>
> - *Relatively small number of layers*. The pruned models used in the skip method are restricted to only 8 layers, which limits their capacity to generate high-quality drafts. This reduced complexity hinders their ability to generate high-quality draft tokens;
> - *Lack of reuse of previous hidden states***.** Unlike our method, which reuses the hidden states from the original LLM for the previous token, the skip method discards this representation and recalculates everything using the pruned LLM.  This previous hidden states from the original LLM, though slightly outdated, contain high-quality information for predicting the next few tokens, being more useful than the newly computed states from a less-powerful, pruned LLM.
>
> This result underscores the importance of reusing hidden states from the original LLM whenever possible. In our experiment, we also compared an augmented version of the skip method that incorporates the hidden states of the original LLM (see Figure 4). While this augmented version does achieve the highest acceptance rate, it comes at the cost of increased latency due to the additional attention layers required, being a suboptimal design.
>
> We hope this explanation clarifies the observed phenomenon and highlights the insights gained from our study.

---

> ### Comment · Reviewer_fPR5 · 2024-12-03
>
> Thank you for providing these additional explanations to my questions. Your responses have shed more light on some aspects of your work, but I still have some concerns and suggestions for improvement:
>
> 1. Regarding model scalability:
>    I appreciate your acknowledgment that the acceleration achieved by your method is likely to decrease as model size increases. I strongly recommend including this discussion in your revised manuscript, along with any preliminary data or theoretical analysis you can provide to support this prediction.
>
> 2. Task type analysis:
>    While this explanation is helpful, I still believe there's room for a more comprehensive analysis. Consider including quantitative data or case studies that demonstrate how different task types benefit from your method to varying degrees. This could significantly strengthen your paper's contributions and practical implications.
>
> 3. Explanation of the skip method results:
>   Thanks for your detailed explanation of why the skip method achieves a lower token acceptance rate compared to your proposed model. The factors you mentioned - the limited number of layers and the lack of reuse of previous hidden states - provide valuable insights.
>
> I believe incorporating these explanations and additional analyses into your paper would significantly strengthen it. In particular, addressing the scalability issue, providing a more in-depth task-type analysis, and explaining the counterintuitive results of the skip method directly in the paper would enhance its contribution to the field.

---

> ### Author Response · Authors · 2024-12-04
>
> Thank you very much for your follow-up feedback!
>
> Regarding 1 and 3, the suggested discussion and modifications have been incorporated into the new manuscript. We would like to express our sincere thanks for these invaluable advices.
>
> Regarding 2, we believe the *acceptance rate gap* between our method and Medusa is a robust quantitative metric, which quantifies how `Markovian' is the dataset i.e. how easily future token can be predicted by ignoring the most recent token. This metric was already presented in our original manuscript and will be discussed in greater details in our revision.
>
> Once again, thank you for the high-quality feedback and insightful suggestions. They really help a lot in enhancing our work.

---

### Official Review · Reviewer_XmjK · 2024-11-04

**Soundness:** 2
**Presentation:** 2
**Contribution:** 2
**Rating:** 5
**Confidence:** 4

**Summary:**

This paper formulates a semi-autoregressive decoding paradigm for LLMs that delegates part of the expensive computation from the original large model to a smaller, more efficient autoregressive model, which allows for substantial reuse of computation in the LLM without missing any token dependencies, thereby striking a good balance between draft quality and drafting speed. Experiments on text summarization, medical QA, code generation, and mathematical reasoning tasks demonstrates the efficacy of our method.

**Strengths:**

- The research direction of this paper, i.e., efficient LLM inference, is meaningful for the development of large language models.
- This paper is well writen and easy for understanding.
- This paper compare several draft methods and combine the advantages of the two previous models, leading to better performance.

**Weaknesses:**

- The main contribution of this article is not clear, which is more likely to be a technical report, thus, I worry about the noverty of this paper.
- Lacking the comparison between current sota speculative decoding methods in main tables.

**Questions:**

see weaknesses.

---

> ### Author Response · Authors · 2024-11-27
> **Clarification on novelty + additional comparison to STOA**
>
> We are grateful for your time and efforts in reviewing our work and the feedback provided. In the text below, a detailed responses to the two mentioned weaknesses are provided. We hope these responses address your concerns effectively.
>
> **On novelty concern of the paper**
>
> We appreciate your feedback and understand that the use of a small autoregressive model as the draft model may appear similar to some recent works, which led to the impression of limited novelty. However, our work is still significantly different from existing works in three senses:
>
> - *Unified framework for draft model design*. While previous works have explored various draft model designs, they have largely treated these approaches in isolation. In contrast, we introduce a novel probabilistic framework that unifies existing designs, allowing for a systematic analysis of the trade-off between draft quality and drafting speed not covered in prior studies (See Figure 3 and Figure 4 in our experiments);
> - *Novel draft model designs with generality and simplicity.* Building on our trade-off analysis, we propose a semi-autoregressive draft model that uses a large model for recent token dependencies and a smaller model for distant dependencies. This unique design allows the use of any off-the-shelf small language models, expanding design spaces and enabling the discoveries of simplified implementations with improved performance.
> - *Specific focus on fine-tuning.* Unlike previous work that focuses on pre-training, we target the underexplored area of fine-tuning for speculative decoding. Fine-tuning presents unique challenges, such as limited data and resources, which require tailored strategies not addressed in prior research.
>
> **On lacking comparison with SOTA methods**:
>
> We thank the reviewer for this constructive feedback. In response, we have added a comparison with the recent work *ReDraft* [r1] — a concurrent submission closely related to one implementation of our proposed semi-autoregressive draft model. This comparison is now included in the main table.
>
> With this addition, our main table now includes comparisons to a diverse set of STOA methods in speculative decoding:
>
> - *Medusa*: A representative method based on multi-token generation.
> - *SkipLayer*: A representative method using an early-exit model as the draft model.
> - *ReDraft*: A representative method using a small autoregressive model as the draft model.
>
> Additionally, our analysis of different implementation of the small language model (SLM) in the proposed semi-autoregressive model also implicitly compares our design to existing SOTA methods in speculative decoding, such as *Hydra* [r2] and *EAGLE* [r3], which are closely related to specific architectural choices for SLM. Please refer to Figure 4 for further details.
>
> *References*
>
> [r1]. Cheng et.al. Recurrent drafter for fast speculative decoding in large language models. arxiv 2403.09919.
>
> [r2]. Ankner et.al. Hydra: Sequentially-dependent draft heads for medusa decoding. COLM 2024.
>
> [r3]. Li et.al. Eagle: Speculative sampling requires rethinking feature uncertainty. ICML 2024.
>
> We hope these clarifications and new results address your concerns. Please let us know if they are satisfactory, and we are keen to hear any further insights from you.

---

### Meta-Review · Area_Chair_AqY1 · 2024-12-25

**Metareview:**

This paper proposes a semi-autoregressive decoding framework to accelerate inference in large language models (LLMs). The main idea is to delegate part of the token generation process to a smaller, cheaper autoregressive model (“drafter”), while relying on the original large model (“validator”) to preserve long-range context and correctness. Empirical evaluations span tasks such as text summarization, medical QA, code generation, and mathematical reasoning, with reported improvements in acceleration and memory cost.

**Strengths** (1) Multiple reviewers mention that the paper is well structured and straightforward to follow. The motivation, algorithm, and experiments are presented in a coherent manner. (2) The paper includes an interesting exploration of simpler models (e.g., one-layer Transformers or RNNs) for generating draft sequences, which demonstrates the potential for reduced computational overhead.

**Weaknesses** (1) Limited novelty: Reviewers note that integrating a small autoregressive model as a “drafter” in speculative decoding has been studied in many prior works. The paper did not acknowledge or compare with the relevant approaches properly. (2) Inadequate Comparisons with Existing SOTA: Multiple reviewers express concern that the paper does not comprehensively compare with state-of-the-art speculative decoding methods.  (3) Some reviewers suggest experiments with larger models to show the scaling and generalizability.

**Decision**
The reviewers broadly agree that while the paper addresses an important problem, its incremental contribution over existing speculative decoding frameworks is not sufficiently demonstrated and not clearly motivated. Consequently, the consensus is that the paper, in its current form, is not ready for acceptance.

**Additional Comments On Reviewer Discussion:**

The authors respond the reviewers' questions, which address the novelty concerns by claiming the proposed method as a more general framework of the semi-autoregressive model, as well as other concerns comparing larger models. Two of the reviewers acknowledge the response from the authors, however, some concerns still remained after the rebuttal, which led to no change on scores

---

### Decision · Program_Chairs · 2025-01-22

Reject